# CoNT: Contrastive Neural Text Generation

**Chenxin An**[1,2]*, **Jiangtao Feng**[2], **Kai Lv**[1], **Lingpeng Kong**[2,3], **Xipeng Qiu**[1], **Xuanjing Huang**[1,4]

[1]Fudan University, [2]Shark-NLP Shanghai AI Laboratory
[3]The University of Hong Kong
[4]Shanghai Collaborative Innovation Center of Intelligent Visual Computing
{cxan20, klv21, xpqiu, xjhuang}@fudan.edu.cn
fengjiangtao@pjlab.org.cn, lpk@cs.hku.hk

## Abstract

Recently, contrastive learning attracts increasing interests in neural text generation as a new solution to alleviate the *exposure bias* problem. It introduces a sequence-level training signal which is crucial to generation tasks that always rely on autoregressive decoding. However, previous methods using contrastive learning in neural text generation usually lead to inferior performance. In this paper, we analyse the underlying reasons and propose a new **Co**ntrastive **N**eural **T**ext generation framework, CoNT. CoNT addresses bottlenecks that prevent contrastive learning from being widely adopted in generation tasks from three aspects – the construction of contrastive examples, the choice of the contrastive loss, and the strategy in decoding. We validate CoNT on five generation tasks with ten benchmarks, including machine translation, summarization, code comment generation, data-to-text generation and commonsense generation. Experimental results show that CoNT clearly outperforms the conventional training framework on all the ten benchmarks with a convincing margin. Especially, CoNT surpasses previous the most competitive contrastive learning method for text generation, by 1.50 BLEU on machine translation and 1.77 ROUGE-1 on summarization, respectively. It achieves new state-of-the-art on summarization, code comment generation (without external data) and data-to-text generation. [2]

## 1 Introduction

Contrastive learning has achieved great success in representation learning [6, 44, 45]. It also attracts enormous interests in neural text generation recently. By creating positive and negative examples in response to unseen (or erroneous) inputs [23], contrastive learning offers a new solution to alleviate the *exposure bias* problem [3, 35] – an autoregressive model trained only using the ground truths exhibits inferior generalization performance. Apart from that, contrastive learning also introduces a sequence-level loss in addition to the conventional token-level language model loss with maximum likelihood estimation (MLE). This is crucial to most conditional text generation tasks (e.g., machine translation and summarization) which are evaluated on sequence-level metrics (e.g., BLEU [32]).

However, it is non-trivial to get contrastive learning working on neural text generation. If we simply use from-batch positive-negative samples following simCLR [6], and adopt the InfoNCE loss [13, 45] which ignores the difference between negative samples (§2.2; Naive CL), the improvement over non-contrastive baselines on generation tasks is rather marginal. Previous work attempts to build better contrastive samples by disturbing the ground truth [10, 23, 30] in the discrete space or the continuous embedding space, but when it comes to text generation tasks, their performance gains are still far from satisfactory.

---

*This work was done during Chenxin An's internship at Shanghai AI Laboratory
[2]The code is available at `https://github.com/Shark-NLP/CoNT`

36th Conference on Neural Information Processing Systems (NeurIPS 2022).

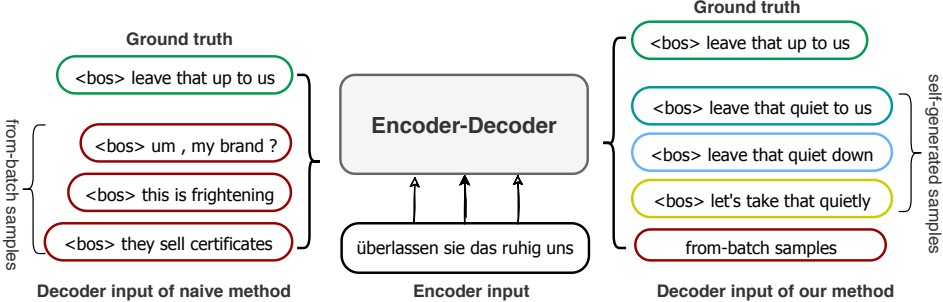

Figure 1: A case study from IWSLT'14 De-En translation task. The naive setting uses from-batch samples following SimCLR [6]. Compared with the naive method, CONT both incorporates self-generated samples and from-batch samples. The border color indicates the acutal distance between the ground truth and the contrastive example.

In this work, we propose a new contrastive neural text generation framework, CONT. CONT does three different things from previous frameworks that make suboptimal use of contrastive learning. First, CONT samples contrastive examples from its own predictions (e.g., through beam search algorithm). This training procedure exposes the model to its mistakes in the inference stage and effectively alleviate the exposure bias problem. We show a comparison between negative samples in CONT and in Naive CL in Figure 1. Second, we use a N-pairs contrastive loss which gives a fine-grained treatment to the contrastive examples based on their sequence-level scores (e.g., BLEU). It allows the model to fully leverage the supervision from the ground truth example (and its own generated examples) to learn a better sequence-level distance function between the source and the target representation. Third, we directly incorporate the learned sequence similarity score from the distance function into the inference stage. This helps the model to find a better global configuration, than merely follows the language model likelihood objective in decoding.

We validate CONT on various important conditional language generation tasks (§4.2), including machine translation, summarization, code comment generation, data-to-text generation, and commonsense generation. Extensive experiments demonstrate that CONT greatly improve the conventional MLE baselines and significantly outperforms all previous contrastive generation models. CONT establishes new state-of-the-art results on summarization, code comment generation (without external data), and data-to-text generation. Particularly, on data-to-text generation and commonsense generation, CONT achieves on-par performance with the powerful large pre-trained models: T5-large, T5-3B [36] with only the base version of T5 while maintaining the efficiency of lightweight models.

## 2 Background

### 2.1 Neural Conditional Text Generation

A neural sequence-to-sequence model [43] $\mathcal{M} = (f, g)$ generates the target sequence conditioning on a source sequence, where $f$ and $g$ denote the encoder and decoder, respectively. It is typically trained using the language model objective with the maximum likelihood estimation (MLE). Given a source sequence $\boldsymbol{x} = \{x_i\}_{i=0}^{M}$ and its target sequence $\boldsymbol{y} = \{y_i\}_{i=0}^{N}$, we minimize the following negative log likelihood (NLL) loss:

$$\mathcal{L}_{\text{NLL}} = -\sum_{t=1}^{N} \log p_\theta(y_t|\boldsymbol{x}, \boldsymbol{y}_{<t}).\tag{1}$$

At training stage, it predict the next word based on previous ground truth input $\boldsymbol{y}_{<t} \in \boldsymbol{y}$, but at inference stage, tokens of $\boldsymbol{y}_{<t}$ are predicted by itself, this introduces the *exposure bias*.

### 2.2 Naive Contrastive Learning for Text Generation

Contrastive text generation introduces a contrastive term in addition to the original NLL loss. In Naive CL, we simply follows SimCLR [6] and use from-batch negative samples $\mathcal{B}$ in the InfoNCE loss [13, 45]:

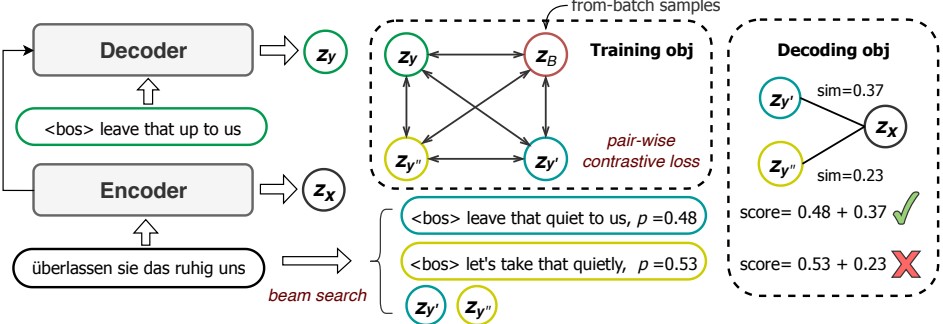

Figure 2: An overview of CoNT. $\mathbf{z}_x, \mathbf{z}_y$ is the representation of source sequence $x$ and its target sequence $y$. $y'$ and $y''$ with their representations $\mathbf{z}_{y'}, \mathbf{z}_{y''}$ are returned by beam search algorithm. The feature representations come from pooling the output of the encoder (source sequence) or decoder (target sequence). Our training objective is obtained by comparing by all contrastive samples in pair. The decoding objective not only considers the likelihood of each sequence, but also the sequence similarity score modeled in training.

$$\mathcal{L}_{\text{NCE}} = -\log \frac{\exp(\cos(\mathbf{z}_x, \mathbf{z}_y)/\tau)}{\sum_{y' \in \mathcal{B}} \exp(\cos(\mathbf{z}_x, \mathbf{z}_{y'})/\tau)}, \tag{2}$$

where $\mathbf{z}_x, \mathbf{z}_y, \mathbf{z}_{y'} \in \mathbb{R}^d$ denote the vector representation of input $x$, ground truth $y$ and negative sample $y' \in \mathcal{B}$, respectively. $\tau$ is the temperature and $\cos(\cdot, \cdot)$ defines the cosine similarity. Intuitively, the contrastive loss $\mathcal{L}_{\text{NCE}}$ seeks to learn a similarity function that drives the distance between the source sequence representation $\mathbf{z}_x$ and its ground-truth target sequence representation $\mathbf{z}_y$ closer.

## 3 Method

In this section, we present our new contrastive neural text generation framework, CoNT . CoNT advances the Naive CL (§2.2) in three aspects. First, CoNT uses negative examples from its own predictions (§3.1) to construct the set $\mathcal{B}$. Second, CoNT replaces the InfoNCE loss (Eq.2) with a N-pairs contrastive loss (Eq.3) which leverages a finer-grained supervision given by the the sequence-level scores of all pairs (§3.2). Third, CoNT incorporates the learned similarity function into its inference score directly (§3.3). An overview of our approach can be found in Figure 2.

### 3.1 Contrastive Examples from Predictions

Instead of only using contrastive examples from the same batch [6], we propose to add new contrastive examples from the model's own predictions. Kalkstein et al. [18] point that using diverse contrastive samples helps the generalization ability of the model. Therefore, we use the diverse beam search algorithm [49] to create contrastive examples from the top-K list of the model's lastest predictions and then append them to the from-batch samples to form the contrastive examples. A warm-up stage where the model is only supervised by $\mathcal{L}_{\text{NLL}}$ is recommended as it guarantees the quality of the examples from the model's prediction. These self-generated contrastive examples alleviate the model's exposure bias. Besides, with the model's performance improving gradually, this approach creates high-quality hard negative examples that is known to be important in contrastive learning [16, 37].

### 3.2 N-Pairs Contrastive Loss

One major drawback of the InfoNCE loss is that it has the same treatment for all negative samples. In text generation, this means that the relative difference between the ground truth and the contrastive examples is ignored, while this can be easily quantified using a sequence level score (e.g. BLEU) and the quality of these contrastive examples varies. To mitigate this problem, we propose to employ a pair-wise margin loss. We first rank all the contrastive examples based on an oracle function $o(\cdot, y)$, which computes a sequence-level score with the ground truth $y$. Given a input sequence $x$, the ground truth $y$, and a set of $K$ contrastive samples $\mathcal{B} = \{y_1, y_2, \cdots, y_K\}$, we can create a series

**Algorithm 1** Inference algorithm: Given an input sequence $\boldsymbol{x}$, a contrastive generation model $\hat{\mathcal{M}} = (\hat{f}, \hat{g})$; return the output sequence.

1: **procedure** BEAMSEARCH($g, H_{\boldsymbol{x}}, b$)  ▷ beam search algorithm
2:  **return** Text, likelihood, logits of the $b$ hypotheses

1: **procedure** INFERENCE($\hat{G}, \boldsymbol{x}$)
2:  $\mathbf{H}_{\boldsymbol{x}} \leftarrow \hat{f}(\boldsymbol{x})$, $b \leftarrow$ beam size, $\alpha \leftarrow$ balance factor $\in (0, 1)$
3:  $\boldsymbol{y}^{1:b}, \mathbf{P}_{\boldsymbol{y}}^{1:b}, \mathbf{H}_{\boldsymbol{y}}^{1:b} = $ BEAMSEARCH($\hat{g}, \mathbf{H}_{\boldsymbol{x}}, b$)  ▷ Get $b$ candidates with beam search
4:  $\mathbf{z}_{\boldsymbol{x}}, \mathbf{z}_{\boldsymbol{y}}^{1:b} \leftarrow$ Avg($\mathbf{H}_{\boldsymbol{x}}$), Avg($\mathbf{H}_{\boldsymbol{y}}^{1:b}$)  ▷ Avg($\cdot$) is an average pooling function
5:  $\mathbf{D}_{\boldsymbol{y}}^{1:b} \leftarrow$ Cosine distance between $\mathbf{z}_{\boldsymbol{x}}$ and representation of hypotheses $\mathbf{z}_{\boldsymbol{y}}^{1:b}$
6:  $\mathbf{P}_{\boldsymbol{y}}^{1:b} \leftarrow$ Likelihood of hypotheses returned by beam search
7:  $k = \arg\max_{i=1..b}\{\alpha * \mathbf{D}_{\boldsymbol{y}}^i + (1 - \alpha) * \mathbf{P}_{\boldsymbol{y}}^i\}$
8:  **return** $\boldsymbol{y}^k$

of example pairs $(\boldsymbol{y}^+, \boldsymbol{y}^-) \in \mathcal{P}$, where $+$ and $-$ are determined by their ranks.[3] The contrastive learning objective is formulated as a margin loss according to their cosine similarity to the source representation $\mathbf{z}_{\boldsymbol{x}}$:

$$\mathcal{L}_{\text{N-Pairs}} = \sum_{(\boldsymbol{y}^+, \boldsymbol{y}^-) \in \mathcal{P}} \mathcal{L}(\boldsymbol{y}^+, \boldsymbol{y}^-) = \sum_{(\boldsymbol{y}^+, \boldsymbol{y}^-) \in \mathcal{P}} \max\{0, \cos(\mathbf{z}_{\boldsymbol{x}}, \mathbf{z}_{\boldsymbol{y}^-}) - \cos(\mathbf{z}_{\boldsymbol{x}}, \mathbf{z}_{\boldsymbol{y}^+}) + \xi\}. \quad (3)$$

We further set $\xi = \gamma * (\text{rank}(\boldsymbol{y}^-) - \text{rank}(\boldsymbol{y}^+))$ following Zhong et al. [57] to reflect the quality difference in these pairs, where $\gamma$ is a hyperparameter controlling the the strength. Full details of the training algorithm can be found in Algorithm 2, Appendix B.

### 3.3 Inference with Learned Similarity Function

Previous inference algorithm for contrastive text generation method [23] usually remains the same with non-contrastive approaches. In CONT, we incorporate the similarity function learned in the N-pairs contrastive loss into the decoding stage. Despite such a inference objective can be generalized to other contrastive learning methods as long as the vector representations for source and target sequence pair exist, the design of CONT can better make use of the learned similarity function (§4.3). The decoding objective in CoNT is to find the sequence $\boldsymbol{y}^*$ that maximizes both the learned similarity score and the conventional language model likelihood:

$$\boldsymbol{y}^* = \arg\max_{\hat{\boldsymbol{y}}}\{\alpha \cdot \cos(\mathbf{z}_{\boldsymbol{x}}, \mathbf{z}_{\hat{\boldsymbol{y}}}) + (1 - \alpha) \prod_{t=0}^{n} p(\hat{y}_t | \boldsymbol{x}, \hat{\boldsymbol{y}}_{<t})\}, \quad (4)$$

where $\mathbf{z}_{\boldsymbol{x}}, \mathbf{z}_{\hat{\boldsymbol{y}}} \in \mathbb{R}^d$ is the vector representation of $\boldsymbol{x}, \hat{y}$, and $\alpha$ is the hyperparameter that balances the contribution of each term. In most cases, $\alpha$ can be directly set to 0.5, tuning $\alpha$ on the validation set will usually get better results. Algorithm 1 illustrates the inference stage in CoNT in details.

The relationship between different modules of CONT is summarized in Figure 3.

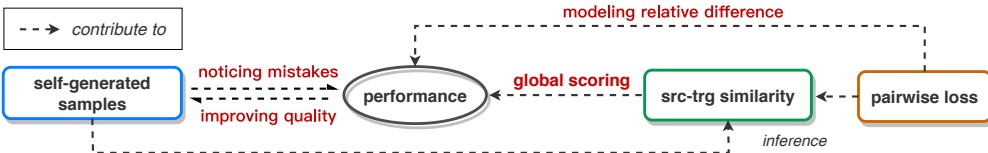

Figure 3: Relationship between different modules in CONT. Both the design of the pairwise loss function and self-generated samples could contribute the source-target similarity function that computes the sequence-level score at inference stage. With the performance improved, self-generated contrastive samples tend to be more indistinguishable.

---

[3]$\mathcal{P}$ contains $C_K^2$ pairs constructed from $\mathcal{B}$, ground truth $\boldsymbol{y}$, and from-batch examples.

# 4 Experiments

We experiment CONT on 5 downstream tasks with 10 different benchmarks. Our contrastive learning framework supports most sequence-to-sequence models at multiple scales. Concretely, we experiment CONT on 4 kinds of base models: (a) Transformer-samll (60M) [47] and transformer-base (220M), (b) T5-small (60M) and T5-base (220M) [36], (c) CodeT5-base (220M) [51] , (d) PEGASUS-large (560M) [55]. Details of our experimental setup for each benchmarks can be found in Appendix C.

On WMT'16 Ro-En (machine translation) and XSum (summarization) which are also used in previous contrastive text generation frameworks [23], results show that CONT is able to substantially improve the MLE baseline and clearly outperform all previous contrastive baselines by a large margin. We also build CONT on state-of-the-art (SOTA) baselines: PEGASUS-large (summarization), CodeT5-base (code comment generation) and achieve new SOTA. Moreover, on data-to-text generation and commonsense generation, CONT also shows its superior performance over strong MLE baselines.

## 4.1 Baselines

1. **Naïve CL [6]**: Naive CL denotes the naive contrastive learning approach that treats the ground truth as the positive sample and the target sequences from the same mini-batch as the negative examples. The training object of naive CL takes the form of Eq. 2. We also implement Naive CL with N-Pairs contrastive loss, it can be viewed as an ablation study when setting beam size of CONT to 0 during training.

2. **SSMBA CL [30]**: Compared with naive CL, SSMBA builds more positive samples via disturbing the ground truth in the discrete space. Concretely, SSMBA first randomly masks 25% tokens in the target sequence and then reconstructs the ground truth with a masked language model BERT.

3. **Dropout CL [10]**: Dropout CL enhances the positive samples by using dropout mechanism on the target sequence. We use the default dropout rate of standard transformer decoder [47] and input the ground truth to the decoder twice.

4. **CLAPS [23]**: CLAPS is previous the best contrastive learning framework for conditional text generation task. In order to provide more challenging contrastive examples, CLAPS propose to simultaneously create extra positive and negative pairs by adding perturbations to the ground truth sequence in the continuous embedding space.

5. **CONT (this work)**: CONT is the contrastive neural text generation framework proposed in this work. We implement its InfoNCE version by treating ground truth as positive sample and self-generated samples are also treated as negative samples.

## 4.2 Quantitative Results

**Machine Translation**   For machine translation, we evaluate CONT on WMT 2016 Romanian-to-English translation task (WMT'16 Ro-En), WMT 2014 English-to-German translation task (WMT'14 En-De) and IWSLT 2014 German-to-English translation task (IWSLT'14 De-En). We use BLEU as the evaluation metric. Results in Table 1 (rows with gray background) indicates our model CONT significantly improves the traditional maximum likelihood estimation training and inference framework. On WMT'16 Ro-En, CONT outperforms previous the best contrastive learning approach CLAPS by 1.50 BLEU and exceeds the MLE baseline by **2.70** BLEU with the same base model T5-small. We also compare the infoNCE loss used in previous methods with the N-Pairs margin loss described in Eq. 3. Results show that the N-pairs contrasting samples generally works better than dividing all samples into predefined positive-negative categories. Similar to CLAPS and Naive CL, only incorporating contrastive learning into training improves the performance of T5-small baseline on WMT'16 Ro-En to 30.55 (+**2.34**) BLEU. If we further add learned target-source similarity as decoding target as Eq. 4, the result is further boosted to 30.91 BLEU. We observe that the benefits of introducing sequence similarity into inference is more obvious on IWSLT'14 De-En – the additional decoding target improves the vanilla beam search algorithm up to 0.86 BLEU.

**Summarization**   For summarization, we use the XSum [28] dataset collected from BBC News whose reference summaries are provided by human writers. We also evaluate CONT on a multi-document summarzation dataset multi-news [9] consisting of news articles from the site newser.com. Compared with the common summarization task, multi-document is more challenging where the model need to automatically summarize several articles and usually has to handle long input sequence

Table 1: BLEU on WMT'16 Ro-En, IWSLT'14 De-En and WMT'14 En-De translation tasks. For IWSLT'14 De-En and WMT'14 En-De, we use Transformer-small (**Tr-small**) and Transformer-base (**Tr-base**) as baselines. For WMT'16 Ro-En, we add a pre-trained baseline **T5-small**. *w/o seq sim* means we use the origin beam search without target-source representation similarity. The best results in each block are underlined and the best results are in bold. Rows in gray denotes the contrastive learning based model strongly outperforms its MLE version. Results with [†] are token from [23].

| Model | WMT'16 Ro-En | | IWSLT'14 De-En | WMT'14 En-De |
|---|---|---|---|---|
| | Tr-small | T5-small | Tr-small | Tr-base |
| MLE | 25.78 | 28.21 | 34.18 | 27.30 |
| *Contrastive loss: InfoNCE loss* | | | | |
| Naive CL | 25.49 | 27.79 | 34.45 | 27.28 |
| SSMBA CL | 25.98 | 28.48 | 34.32 | 27.16 |
| Dropout CL | 26.01 | 29.10 | 34.41 | 27.34 |
| CLAPS[†] | 23.59 | 29.41 | – | – |
| CoNT | 25.74 | 29.64 | 34.46 | 27.35 |
| *Contrastive loss: N-Pairs loss* | | | | |
| Naive CL | 26.15 | 29.86 | 34.47 | 27.41 |
| *w/o seq sim* | 26.27 | 29.74 | 34.26 | 27.45 |
| CoNT | **27.70** | **30.91** | **35.55** | **28.04** |
| *w/o seq sim* | 27.42 | 30.54 | 34.69 | 27.77 |

Table 2: ROUGE score on Summarization datasets. Results with [†] are token from [23] and results with [*] are from [55]. Current state-of-the-art models and the best results are in bold. Previous SOTA means the best results before CoNT.

| Model | XSum | | | Multi-News | | |
|---|---|---|---|---|---|---|
| | R-1 | R-2 | R-L | R-1 | R-2 | R-L |
| T5-small | 36.10 | 14.72 | 29.16 | 42.36 | 15.34 | 21.91 |
| T5-SSMBA CL | 36.58 | 14.81 | 29.68 | 42.06 | 14.98 | 21.73 |
| T5-Dropout CL | 36.82 | 14.93 | 29.26 | 42.43 | 15.32 | 21.95 |
| T5-CLAPS[†] | 37.89 | 15.78 | 30.59 | – | – | – |
| T5-Naive CL | 36.34 | 14.81 | 29.41 | 42.20 | 15.18 | 21.78 |
| T5-Naive CL (N-Pairs) | 37.76 | 15.48 | 30.15 | 43.04 | 15.83 | 22.03 |
| T5-CoNT | 39.66 | 16.96 | 31.86 | 44.08 | 16.39 | 22.58 |
| Previous SOTA[*] | 47.61 | 24.57 | 39.44 | 47.52 | 18.72 | 24.91 |
| PEGASUS (base)[*] | 39.79 | 16.58 | 31.70 | 42.24 | 13.27 | 21.44 |
| **PEGASUS (large)**[*] | 47.21 | 24.56 | 39.25 | 47.52 | 18.72 | **24.91** |
| PEGA-CoNT | **47.76** | **24.69** | **39.46** | **48.68** | **19.29** | 24.58 |

and target sequence. Experimental results are in Table 2. The first block includes the performance of different contrastive frameworks with T5-small. On XSum, it shows that our proposed model strongly outperform previous contrastive frameworks by about 2.0 ROUGE-1 score. We also illustrate our method is not restricted to the small model. By employing CoNT on state-of-the-art base model PEGASUS, it is able to establish new state-of-the-art on the two summarization benchmarks.

**Code Comment Generation**    Code comment generation aims to generate an English description for a function-level code snippet. We test our method on two widely used datasets Java and Python from the CodeXGLUE benchmark [27]. Results are shown in Table 3. Our model is built upon state-of-the-art pre-trained model on program language model CodeT5-base. CodeT5-Dual-Gen means they further involve a comment-to-code task which is the best model on Python and Java without using external data. We also include the results of earlier strong pre-trained baselines: PLBART

Table 3: BLEU on two code comment generation datasets Java and Python. Results with † and * are from [51]

| Model | Python | Java |
|---|---|---|
| CodeBERT † | 19.06 | 17.65 |
| PLBART † | 19.30 | 18.45 |
| CodeT5 † | 20.01 | 20.31 |
| **CodeT5-Dual-Gen**† | 20.11 | 20.41 |
| **With N-Pairs CL** | | |
| CodeT5-Naive CL | 20.26 | 20.31 |
| CodeT5-CONT | 20.43 | 20.56 |
| **With External Training Data** | | |
| CodeT5-Multi-Task† | 20.36 | 20.46 |
| **REDCODER*** | **21.01** | **22.94** |

Table 4: BLEU on data-to-text generation dataset WikiBio. We run our model three times and report the mean and variance of the BLEU metric. Results with † are token from [25] and results with * are from [2].

| Model | BLEU |
|---|---|
| Table NLM † | 34.70 ±0.36 |
| vanilla Seq2Seq † | 42.06 ±0.32 |
| StructureAware † | 44.89±0.33 |
| **R2D2** * | 46.23±0.15 |
| T5-small † | 46.02±0.36 |
| **With N-Pairs CL** | |
| T5-small-Naive CL | 46.50±0.24 |
| T5-small-CONT | **47.17**±0.19 |

Table 5: Results on the dev set and test set Totto. PAR is short for PARENT score. Dev Set (Non) means the non-overlap subset of the dev set. results with † are reported in [17].

| Model | Dev Set (All) | | Dev Set (Non) | | Test Set (All) | | |
|---|---|---|---|---|---|---|---|
| | BLEU | PAR | BLEU | PAR | BLEU | PAR | BLEURT |
| BERT-to-BERT† | 44.0 | 52.6 | 34.8 | 46.7 | 44.0 | 52.6 | 0.121 |
| T5-large† | 48.1 | 57.3 | 39.8 | 52.8 | – | – | – |
| **T5-3B**† | 48.4 | 57.8 | 40.4 | 53.3 | **49.5** | 58.4 | 0.230 |
| T5-base† | 47.7 | 57.1 | 39.6 | 52.6 | – | – | – |
| T5-base-CONT | **49.2** | **59.4** | **41.5** | **55.0** | 49.1 | **58.9** | **0.238** |

and CodeBERT. We also report some data augmentation methods in the third block of Table 3. CodeT5-Multi-Task makes use of training datasets of other program languages. REDCODER [34] uses retrieval to enhance the task with open-source code base and achieve the bset results on this task. Our model is orthogonal to these methods and clearly outperforms all baselines without external data.

**Data-to-text Generation** Data-to-text generation aims to produce text from non-linguistic input. The first benchmark we use is WikiBio [22] consisting of biography pairs from English Wikipedia where the infobox is treated as input sequence, and the target sequence is the first sentence of the biography. Totto [33] is also collected from Wikipedia whose input is a table with its highlighted cells and target sequences are professionally annotated by human. Results on WikiBio is shown in Table 4, the performance of some popular baselines (first block) are token from [25]. R2D2 [2], using XLNet [53]-large as base model, is previous state-of-the-art model on WikiBio. We experiment CONT on T5-small, and results show that we exceed R2D2 by about 0.94 BLEU. The test set and dev set of Totto are both split into two parts - overlap and non-overlap. The non-overlap part contains out-of-domain samples from the training set. The test set of Totto is invisible and we report the results on dev set and test set by the feedback of the Totto authors. We also add PARENT [8] and BLEURT [40] as evaluation metrics. PARENT is a word-overlap based metric that designed to evaluate the factual accuracy of generation results. BLEURT is trained under human supervision and correlates well with human judgement. As we can see from Table 5, by comparing our model with different T5 variants, we show that CONT is able to greatly outperform the large version of T5 even built on a T5-base model. Compared with previous state-of-the-art model T5-3B, our model still show advantage in PARENT and BLEURT.

**Commonsense Generation** The task of commonsense generation aims to explicitly test the ability of machines on commonsense reasoning. The source sequence consists of a set of concepts and the target sequence is a fluent sentence mentioning all the input concepts. We evaluate CONT

Table 6: Results on CommonGen. Results with [†] are reported in [24]. The metrics used in the official leaderboard are in bold. Human performance is also reported as an upper bound.

| Model | ROUGE-2/L | | BLEU-3/**4** | | METEOR | **CIDEr** | **SPICE** | Coverage |
|---|---|---|---|---|---|---|---|---|
| GPT-2[†] | 16.85 | 39.01 | 33.92 | 23.73 | 26.83 | 12.19 | 23.57 | 79.09 |
| BART[†] | **22.02** | 41.78 | 39.52 | 29.01 | 31.83 | 13.98 | 28.00 | **97.35** |
| T5-large[†] | 21.74 | 42.75 | **43.01** | **31.96** | 31.12 | 15.13 | 28.86 | 92.29 |
| T5-base[†] | 14.63 | 34.56 | 28.76 | 18.54 | 23.94 | 9.40 | 19.87 | 76.67 |
| T5-base-CoNT | 20.96 | **43.15** | 42.60 | 31.42 | **32.05** | **15.96** | **28.95** | 96.55 |
| Human | 36.72 | 54.45 | 52.55 | 46.49 | 38.79 | 37.64 | 52.43 | 99.33 |

Table 7: BLEURT and BERTScore on the test set of 4 translation and summarization datasets. The first column of each dataset represents BLURT and the second column is BERTScore.

| Model | **IWSLT'14 De-En** | | **WMT'16 Ro-En** | | **XSum** | | **Multi-News** | |
|---|---|---|---|---|---|---|---|---|
| MLE | 0.137 | 62.28 | 0.272 | 69.09 | -0.552 | 44.10 | -0.568 | 17.21 |
| CoNT | 0.167 | 63.38 | 0.281 | 69.33 | -0.462 | 46.78 | -0.505 | 17.47 |

on the CommonGen [24] benchmark with a hidden test set and the final results is obtained with the help of the authors of CommonGen. In addition to the mostly used metrics, CIDEr [48] and SPICE [1], concerning to evaluating semantic faithfulness, are highlighted by the leaderboard of commonGen. In Table 6, we demonstrate that the lightweight T5-base model is able to greatly benefit from our contrastive learning framework. Moreover, CoNT not only surprisingly outperforms its MLE baseline but also surpass the large version of T5 in terms of CIDEr and SPICE metrics.

**Advanced Evaluation Metrics**    Considering the training efficiency of CoNT, we mainly select the lexical matching metrics as oracle measurement function. To verify that the improvement brought by CoNT is not due to the over-fitting of lexical matching metrics, we further evaluate generated text with advanced metrics based on neural models: BERTScore [56] and BLEURT [40]. For BERTScore, we use their roberta-large_L17_no-idf_version and for BLEURT we use the default setting provided on their github[4]. Results are shown in Table 7. The base model used on IWSLT'14 De-En is transformer small and on the other datasets we select T5 as the base model. For all datasets CoNT also make non-trivial improvements in terms of the two neural metrics. Particularly, CoNT improve the results of MLE model on IWSLT'14 De-En by 0.03 BLEURT and improve the results of MLE model on XSum by 2.68 BERTScore.

To get more accurate and convincing results, we also conduct a ranking based human evaluation on two mainstream tasks: machine translation (IWSLT'14 De-En) and text summarization (XSum) with 60 samples for each tasks. Following Cheng and Lapata [7], we hired 2 annotators asking them to rank the given candidate output based on fluency, coherence, and their personal preference ( rank these systems 1st, 2nd, and so on) and we calculate the average ranking. For each sample, there are four candidates consist of a human-written reference, a sequence from MLE model, a sequence from Naive CL, and a sequence from CoNT. Table 8 shows the results of our human evaluation. Generally CoNT outperform all baseline systems according to the average ranking.

## 4.3 Discussion

**Discrimination of Hard Negative Samples**    To deeply look into the learnt representations, we visualize target sequences, that is trained by MLE, naive CL and CoNT on IWSLT'14 De-En, with the t-SNE algorithm [46]. The visualized sequences consist of three groups of target sequence: a) batch targets that is mostly unrelated to ground -truth target; b) beam search hypothesis that could be of high/low quality; c) ground truth target. As can be seen in Figure 4a, the representations trained by MLE are distributed almost uniformly in the vector space, and there are no clear boundary between one group to another. With naive CL, we find there is clear boundary between batch tokens and

---

[4]https://github.com/google-research/bleurt

Table 8: Results of human evaluation on the test sets of translation and summarization.

| Model | Machine translation | | | | | Summarization | | | | |
|---|---|---|---|---|---|---|---|---|---|---|
| | 1st | 2nd | 3rd | 4th | **avg rank** | 1st | 2nd | 3rd | 4th | **avg rank** |
| Ground truth | 0.88 | 0.08 | 0.0 | 0.04 | 1.2 | 0.52 | 0.25 | 0.12 | 0.12 | 1.86 |
| CoNT | 0.07 | 0.5 | 0.31 | 0.12 | 2.48 | 0.27 | 0.4 | 0.2 | 0.13 | 2.2 |
| Naive CL | 0.02 | 0.25 | 0.4 | 0.33 | 3.04 | 0.12 | 0.22 | 0.38 | 0.28 | 2.82 |
| MLE | 0.03 | 0.17 | 0.28 | 0.52 | 3.29 | 0.1 | 0.13 | 0.3 | 0.47 | 3.13 |

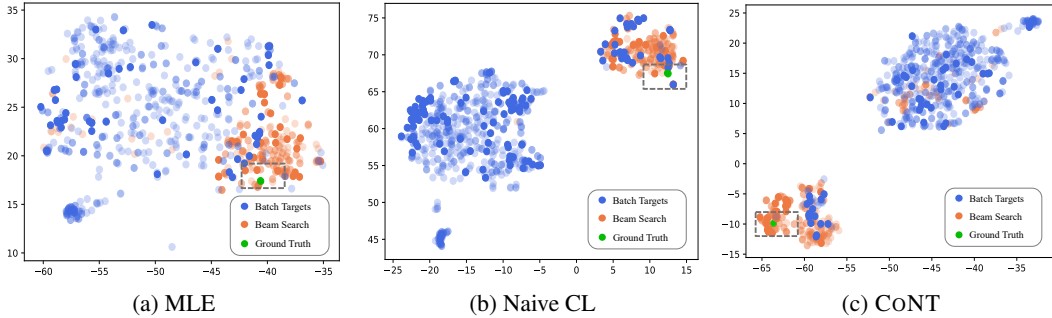

(a) MLE      (b) Naive CL      (c) CoNT

Figure 4: T-SNE experiments on IWSLT'14 De-En. Each point represents a target sequence. Batch target is blue; beam search hypothesis is orange; ground-truth sequence is green. Darker points indicate sequences with higher BLEU.

others. Naive CL does contribute to discriminating related sequences with unrelated ones, but it still cannot distinguish hypothesis of high quality from the others. Even without contrastive learning, the generation model trained has already pulled from-batch samples away from the ground truth and the naive contrastive learning procedure is only to make the margin more obvious. As for CoNT, a set of hypotheses of low quality are excluded from the neighborhoods of the ground-truth targets. The experimental results verifies that CoNT enables better representations in sequence generation.

**Sequence Likelihood and Sequence Similarity** We also perform ablation study on our ranking objective and self-generated negative samples on IWSLT'14 De-En with Transformer-small as base model. For the ranking objective, we compare our N-pairs contrastive objective with InfoNCE with increasing $\alpha$. Figure 5a that N-pairs contrastive loss consistently outperform InfoNCE. With $\alpha \in [0.3, 0.7]$, N-pairs contrastive loss can benefit target-source representation similarity, a sequence-level score in inference process, while InfoNCE cannot. For the sampling strategy, we compare CoNT with naive CL and vanilla MLE. CoNT use self-generated hypotheses as negatives while

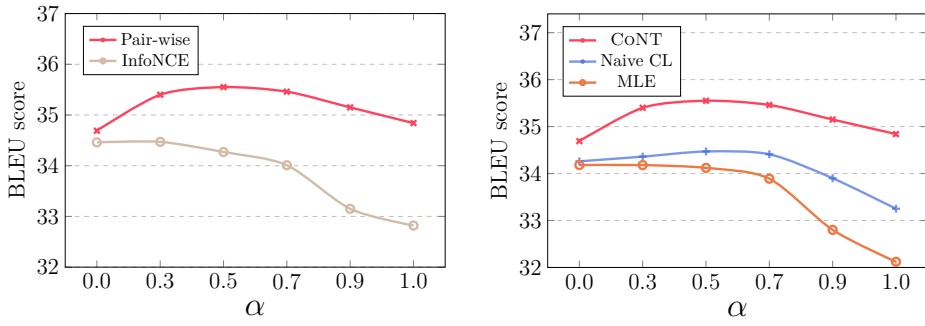

(a) Relationship between $\alpha$ and BLEU with different contrastive loss.      (b) Relationship between $\alpha$ and BLEU with different training methods.

Figure 5: Ablation study on the balance factor $\alpha$ on the test set of IWSLT'14 De-En where $\alpha = 0.0$ means selecting output only relying on likelihood and $\alpha = 1.0$ means choosing output with only sequence similarity.

naive CL only use samples within the same mini-batch. Figure 5b shows that contrastive learning with self-generated hypotheses is more effective than using batch samples. With $\alpha \in [0.3, 0.7]$, CONT gains about $1.0$ BLEU improvements, while naive CL is only improved with less than $0.5$ BLEU.

## 5 Related Work

**Contrastive Learning**  Contrastive learning [6, 13, 39] aims to learn a better representation via contrasting positive and negative samples. It also has been widely used in the field of natural language processing [10, 14, 20, 21, 59]. Jiang et al. [15] show that contrastive learning helps learn a robust pre-trained model and Lee et al. [23] first introduce contrastive learning into text generation to mitigate the exposure bias problem. They propose an adversarial method to build more challenging positive-negative samples in additional to the from-batch samples. SimCTG [42] is also a contrastive framework on text generation. Our work differs from SimCTG from motivation and method. They introduce contrastive leanrning mainly to encourage diversity which is important in dialogue systems. And they perform token-level contrastive learning while our method focus on sequence-level contrastive examples.

Adopting binary supervision in contrastive loss is originally proposed in FaceNet [39] which helps learn the face recognition of the same person in various positions and angles. Given an anchor face image $\mathbf{x}$, a positive sample $\mathbf{x}^+$ (usually the same person) and a negative sample $\mathbf{x}^-$ from other people, the triplet loss makes $\mathbf{x}^+$ become close to $\mathbf{x}$ and maximize the distance between $\mathbf{x}$ and $\mathbf{x}^-$. The pair-wise contrastive loss has also been widely used in metric learning Chen and Deng [5], Kim et al. [19], Wang et al. [50]. Sohn [41] extend the triplet loss to multi-class and multi-pair. Recent work thinks the margin value between samples should not be fixed. Zhou et al. [58] divide the sample set into multiple subsets and assign different margin value to different subsets. Ha and Blanz [12], Zhong et al. [57] suggests dynamically adjust the margin value via a determination metric.

**Post-generation Re-ranking Models**  Post-generation re-ranking re-score the multiple output sequences via training another re-ranking module. Noisy Channel Modeling (NCM) [29, 54] is a widely-used re-ranking scheme for neural machine translation. NCM parameterizes the noisy channel probability with a sequence-to-sequence model. There also various structures to instantiate the re-ranking module: Gulcehre et al. [11] select the candidate with a language model, Bhattacharyya et al. [4] leverage an energy-based model in NMT and Liu and Liu [26], Salazar et al. [38] re-score candidates with masked language models such as BERT. Despite this paradigm achieves impressive results while having a large size of candidate sequences, most of post-generation re-ranking systems trade efficiency and simplicity for accuracy.

## 6 Conclusion

We introduce a new contrastive neural text generation framework called CONT. It models an additional contrastive learning objective to provide a sequence-level supervision for auto-regressive neural text generation models. We explore three shortcomings that limit the development of contrastive learning on text generation tasks. Results on five generation tasks with ten different benchmarks show that CONT not only clearly beats all previous contrastive generation models, but also boosts the performance of state-of-the-art large models to a new level. CONT practically does not have a negative impact on decoding speed. Nevertheless, CONT suffers from the training inefficiency problem. In general, the total training time of CONT is about $2{\sim}4$ times more than that of a MLE based model. A detailed discussion and some speed-accuracy trade-off tricks can be found in Appendix B. Speeding up the training stage without losing accuracy is the next important step to improve CONT.

## Acknowledgement

We would like to thank the anonymous reviewers for their valuable advice. This research was supported in part by the joint research scheme of the National Natural Science Foundation of China (NSFC) and the Research Grants Council (RGC) under grant number N HKU714/21.

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
