# OpenReview forum: "CoNT: Contrastive Neural Text Generation"
_NeurIPS.cc/2022/Conference — NeurIPS 2022 Accept_

### Official Review · Reviewer_Mnn7 · 2022-07-03

**Rating:** 5
**Confidence:** 4
**Soundness:** 2 fair
**Presentation:** 4 excellent
**Contribution:** 3 good

**Summary:**

This paper presents a new algorithm for training neural text generation models. The standard approach to training text generators uses the MLE objective, which is known to suffer from the "exposure bias" problem --- a mismatch between training and testing. This paper proposes to tackle this issue using a contrastive learning objective.

The authors perform contrastive learning on the encoder / decoder representations of the model, pushing the input vectors close to the ground truth and away from negatives. Three key ideas are used to implement this --- (1) model generations are used to create harder negatives than simply in-batch negatives; (2) negatives are broken down into (positive, negative) pairs according to an oracle scoring function (like BLEU); (3) during decoding an additional re-ranking step is used which minimizes the cosine similarity of the learned contrastive representations.

The authors evaluate their approach on 5 conditional text generation tasks (over 10 datasets), and report 0.5-2 point improvements across tasks over competing baselines using automatic token-overlap metrics.

**Questions:**

What are the GPU memory requirements for the approach?

Does performance improve with more negative samples?

how important is beam size during inference? I'm guessing larger beam-sizes are important to have more hypotheses to re-rank with the vector cosine similarity.

**Limitations:**

Yes, good limitations section!

**Strengths And Weaknesses:**

**Strengths**

1. The authors present a novel method of incorporating contrastive learning into training text generation models. The method seems like a neat way of tackling the exposure bias problem in text generation since it's a sequence level objective (rather than MLE).

2. The authors have a comprehensive experimental setup covering 10 datasets across five practically relevant text generation tasks (like summarization, translation). The authors consistently report small improvements in automatic metrics over competing baselines across all tasks, despite the different tasks having very different input/output spaces.

3. Experimental results are complemented with good ablation studies justifying each design choice, and model generations in the appendix.

4. The authors have a good limitations section acknowleding the slower training time of the approach along with strategies to reduce its effect.

**Weaknesses**

I have two major concerns with the paper:

1. *No human evaluation*: While experiments were conducted on 10 benchmarks, no human evaluation was done --- all improvements are solely on automatic evaluation metrics. Most of the metrics used are token overlap metrics well known to be brittle, see [1, 2, 3, 4, 13, 14] for example. I think using human evaluation (for a subset of tasks at least) is particularly important in this paper due to the oracle scoring function used during training. What oracle function was used in different tasks? Is it the same metric that is used for evaluation of the models? My worry is that the oracle function will encourage over-fitting to the properties of the flawed metric (like BLEU / ROUGE) rather than actually improve generated text. Also, why not use BLEURT for translation evaluation [4], or some of the newer summarization metrics [5] with higher correlation that ROUGE? In any case, I think human evaluation is a gold standard which needs to be done in text generation work to validate the methods.

2. *Scaling the idea*: The authors primarily try their idea on small transformer models (60M - 220M parameters), with PEGASUS-large (560M) for XSUM being an exception. While it's great that contrastive learning on smaller models matches the performance on bigger models, I'm curious to know whether the proposed approach scales to larger transformer models like T5-large, T5-3B and T5-11B (I think T5-large / 3B can be done on NVIDIA A100 GPUs with Adafactor?). This is important for a few reasons --- (1) the proposed approach has a 2-4x slowdown in training (as the authors discuss in their limitations section), so it may be more efficient to simply scale the model size with MLE (and get similar performance); (2) scaling could go either way in terms of performance improvements, and this is important factor determining the generality of the approach (informal tweet on this in [7]) (3) some of the results (like IWSLT De-En) are below SoTA [6], and the CONT approach may be able to cover up the gap using larger Transformers.

**Minor**

1. It will be good to cite and discuss the relations to [8, 9] in the related work since they use contrastive learning for text generation as well. These two papers have come out in the last 2-3 months (contemporary to this paper), so no direct comparison is necessary. A few older papers with related ideas you may want to cite as well --- [10, 11, 12].

**Overall**: This paper has good clarity and moderate originality, quality and significance. Overall, I liked the ideas in the work and the comprehensive evaluation setup across 10 tasks, but the lack of human evaluation + small scale of the most models tested (60M - 220M parameters) makes me slightly lean reject. The missing human evaluation is a bigger concern for me. I will be happy to move to the accept range with human evaluation on at least 2 of the 10 datasets, or a strong rebuttal discussing why the human evaluation / model scaling is not needed.

---

**After Rebuttal** - The authors presented a preliminary human evaluation with promising results. I am raising my score to 5, but I recommend the authors to conduct a more thorough human evaluation in the next version of the paper.

[1] - https://arxiv.org/pdf/2202.06935.pdf
[2] - https://arxiv.org/abs/1604.00400
[3] - https://arxiv.org/abs/2103.06332
[4] - https://aclanthology.org/2020.acl-main.704/
[5] - https://arxiv.org/abs/2111.09525
[6] - https://paperswithcode.com/sota/machine-translation-on-iwslt2014-german
[7] - https://mobile.twitter.com/_jasonwei/status/1526589104758042624
[8] - https://arxiv.org/abs/2205.09726
[9] - https://arxiv.org/abs/2202.06417
[10] - https://arxiv.org/abs/2004.11714
[11] - https://aclanthology.org/D16-1137/
[12] - https://arxiv.org/abs/1804.06451
[13] - https://arxiv.org/pdf/2107.10821.pdf
[14] - https://arxiv.org/abs/2006.06264

---

> ### Author Response · Authors · 2022-07-31
> **Response to Reviewer Mnn7**
>
> Thank you for a very detailed review. Using BLEURT or other metrics like BERTScore as the oracle function might benefit the accuracy, but  they will dramatically slow down the training speed.  Since we chose the token overlap metric as oracle measurement, we agree that a human evaluation is an informative complement for the paper. We have conducted a preliminary human evaluation on IWSLT-14 En-De (translation) and XSum (summarization) with 30 samples for each datasets.  We hired 2 annotators asking them to rank 4 given candidate output based on fluency, coherence, and their personal preference. The 4 candidates consist of a human-written reference, a sequence from MLE model, a sequence from Naive CL, and a sequence from CoNT. We will add more comprehensive human evaluations to our camera-ready version.
>
> > *Results from participant #1  -- machine translation*
>
> | Methods | 1st | 2nd | 3rd | 4th | Avg rank|
> | :-----| :----: | :----: | :----: | :----: | :----: |
> | CoNT | 0% | 60% | 26.7% | 13.3%  | 2.5 |
> | Naive CL | 0% | 33.3% | 36.7% | 30% | 3.0 |
> | MLE | 0% | 6.6%| 36.7% | 56.7% | 3.5 |
> | Ground truth | 100% | 0% | 0% | 0% | 1.0 |
>
> > *Results from participant #1  -- Summarization*
>
> | Methods | 1st | 2nd | 3rd | 4th | Avg rank|
> | :-----| :----: | :----: | :----: | :----: | :----: |
> | CoNT | 26.7% | 43.3% | 23.3% | 6.7%  | 2.1 |
> | Naive CL | 13.3% | 33.3% | 36.7% | 16.7% | 2.6 |
> | MLE | 0% | 3.3%| 33% | 63% | 3.6 |
> | Ground truth | 60% | 20% | 6.7% | 13.3% | 1.7 |
>
> > *Results from participant #2  -- machine translation*
>
> | Methods | 1st | 2nd | 3rd | 4th | Avg rank|
> | :-----| :----: | :----: | :----: | :----: | :----: |
> | CoNT | 6.6% | 46.7% | 36.7% | 10%  | 2.5 |
> | Naive CL | 0% | 43.3% | 26.7% | 30% | 2.9 |
> | MLE | 3.3%| 6.7% | 36.7% | 53.3% | 3.4 |
> | Ground truth | 90% | 3.3% | 0% | 6.7% | 1.2 |
>
> >*Results from participant #2  -- Summarization*
>
> | Methods | 1st | 2nd | 3rd | 4th | Avg rank|
> | :-----| :----: | :----: | :----: | :----: | :----: |
> | CoNT | 36.7% | 36.7% | 13.3% | 13.3%  | 2.1 |
> | Naive CL | 6.7% | 30% | 43.3% | 20% | 2.7 |
> | MLE | 0% | 3.3%| 33% | 63% | 3.3 |
> | Ground truth | 53.3% | 23.3% | 6.7% | 16.7% | 1.9 |
>
> ----
> **Below we address your other concerns one-by-one.**
>
> >Q1:  Try large-version transformer models for CoNT.
>
> In real-world applications, text generation is often deployed as a service that handles thousands of requests per hour. The main advantage of base models + CoNT is its inference efficiency, including less floating point operations (means less energy needed for each forward) and faster inference speed. We also agree that scaling to the larger model will help prove the effectiveness of CoNT. Therefore, we validated CoNT on the  strong summarization model PEGASUS-large and achieved new SOTA. Scaling CoNT to large transformers on all tasks is still unaffordable with 4 A100 GPUs.
>
> >Q2: What are the GPU memory requirements for the approach?
>
> For IWSLT14 De-En, when the number of samples is set to 16 and max-tokens is set to 4096 (maximum number of tokens in a batch), the maximum GPU memory consumption is ~22GB.  For comparison, under the same environment , the naive approach needs ~20GB GPU memory and the MLE model needs ~7 GB GPU memory.  With gradient accumulation=4  and max-tokens=1024, CoNT can run on a 12GB TITAN Xp GPU.  Considering the training time, we recommend using 8 RTX 3090 GPUs or 2 A100 GPUs  for training.
>
> The contrastive learning based methods generally take more GPU memory because the decoder has to be exposed to both valid and erroneous input.  Notice that CoNT does not need large batch to enrich the negative samples allowing gradient accumulation to simulate large batch size.
>
> >Q3: Does performance improve with more negative samples?
>
> Yes,  if we keep increasing the number of contrastive examples, performance will be further boosted.
> On IWSLT14 de-en,  when the number of negative samples are increasing: 8 => 16  => 32, its BLEU are gradually improved : 34.73 => 35.27 => 35.55. We did not further enlarge the example size due to the increasing computation cost.
>
> >Q4: how important is beam size during inference? I'm guessing larger beam-sizes are important to have more hypotheses to re-rank with the vector cosine similarity.
>
> In our experiments, we suggest setting the beam size to 6,8,12 (details of the hyperparamters during inference can be found in Table  8 in Appendix). We found further increasing the beam size also  reduces performance.  As can be seen in figure 5, performance drops when we rely entirely on vector cosine similarity and we also need to incorporate the likelihood.  Usually, using a large beam size will harm performance for most MLE models and CoNT as well, despite it being more robust to that as MLE only contributes to part of the loss in CoNT.
>
> >Q5: Minor.
>
> We will discuss the work [8, 9, 10, 11, 12] that you mentioned in our related work section. Thank you for your suggestions that would make our related work section more thorough.

---

> > ### Comment · Reviewer_Mnn7 · 2022-08-07
> > **Thank you, score raised to 5**
> >
> > Thank you for your detailed reply and good to see promising initial results in the human evaluation. I've raised my score to 5, but I recommend a more thorough human evaluation in the next version of the paper.
> >
> > Also, I agree that BLEURT / BERTScore is slow during training. What I meant was, use it at inference time only to judge the final quality of outputs. They have good correlation with human judgments + it removes the confound of identical scoring functions for oracle / evaluation.

---

> > > ### Author Response · Authors · 2022-08-08
> > > **Response to Reviewer Mnn7 Comments**
> > >
> > > Thanks for your feedback and thanks again for  the time and efforts you provided!  Based on your comments, we will add a more  thorough human evaluation and a new section named **Advanced Evaluation Metrics** in our experiments part  where we will further validate CoNT  with metrics like BLEURT, BERTScore.

---

### Official Review · Reviewer_EAev · 2022-07-08

**Rating:** 6
**Confidence:** 3
**Soundness:** 3 good
**Presentation:** 3 good
**Contribution:** 3 good

**Summary:**

The paper presents a new contrastive neural text generation framework, CoNT. There are three major technical improvement and contribution for this framework. First, they use model’s beam search result as additional contrastive examples. Second, they construct an N-pairs contrastive loss, compared to traditional pairwise loss. Third, they manipulate the hypothesis score by adding the learned similarity function. The experiment on five downstream tasks shows that the proposed methods are effective in general.


**Questions:**

The diverse beam search algorithm might not be actually diverse given my experience. Have you considered using some other methods, like sampling methods? Top-p sampling, top-k sampling, something like these.

The arrows between z_y, z_b, … in Figure 2 are a little confusing. The use of zy’ and zy’’ is also a little confusing. Please add some notation and explanation for these arrows and symbols. Also, some lines are used to connect z_y and z_x while some are for z_y and z_y’’. The function of lines and arrows needs to clearly stated.


**Strengths And Weaknesses:**

The approaches proposed in this paper are overall intuitive and effective. The use of model’s prediction is a smart way to collect more high quality negative examples. The n-pairs cross constrictive loss captures the relative difference between ground truth and the constructive examples. The approach is pretty general and could be applied to many tasks.

The experiment is comprehensive. It includes experimental results and analysis on 5 downstream tasks and 10 different tasks. Although some of the potential comparison is not provided, the results generally look good to me. The extensive comparison will be beneficial to the community.

The paper is well written and presented. I don’t have any major difficulty understanding the paper.


Weakness

The results aren’t 100% solid to me. According the Liu and Liu [24] cited by this paper, on XSUM, one of the summarization dataset, BART/PEGASUS achieve 45.14 and 47.21 on ROUGE-1. The contrastive learning method proposed by Liu and Liu [24], SimCLS, achieves 47.61 on R-1. Given the paper cites [24], are there any reason 1) the “earlier SOTA” was much lower than baseline methods in previous work (SimCLS is 47.61)? 2) the SimCLS is not directly listed and compared?
I have experience working on text summarization, so It’s easier for me to verify the results in summarization. I am a little worried since the author shows many crappy baselines “T5-xxx”, which seems to be misleading.


There are some mathematical issue about some of the equations. In Eq 4, the likelihood is defined as the product of the decoding sequence. A product of many probabilities will be a very small number, right? In that sense, the second term in the equation will very small compared to the first term ranging from -1 to 1. The equation is just wrong to me. Also, I don’t see the reason using alpha and 1-alpha to balance these two terms because they aren’t in the same range or scale. In Figure 2, the prob of two examples are -.48 and 0.53, which don’t seem to be the product of a sequence of probabilities.
In this sense, the result of Figure 5 seems spurious to me.

---

> ### Author Response · Authors · 2022-07-31
> **Response to Reviewer EAev**
>
> Thank you for your valuable comments that will  help make our article clearer,  especially on text summarization. We summarize your main concerns as follows:
>
> ----
> >Q1: Why most contrastive baselines are built on T5?
>
> Actually, we agree that T5 is not the strongest baseline in text summarization. This is also one of the reasons that we further conduct experiments of CoNT on PEGASUS-large. Nevertheless, T5 should be deemed as an important baseline as previous contrastive text generation methods were built on T5. The experiments on T5 can demonstrate the effectiveness of our contrastive approach including contrastive examples construction and loss function, under the same setting as previous methods.
>
> >Q2:  Misunderstanding of "Earlier SOTA"  in table 2.
>
> Thank you for the very valuable advice. We explained that the "Earlier SOTA" in the caption of Table 2 is "the best results **before PEGASUS**" and it was directly taken from their paper.  But "Earlier SOTA" in this paper may cause some ambiguity so that we have changed the  "Earlier SOTA" in table 2  "Previous SOTA" which means the best results **before CoNT** (not PEGASUS).
>
> **Details:**
>
> |       |          |  XSum |        |    |           |  Multi-News  | |
> | :-----| :-----:| :----: | :----:  | :-:|  :---:  | :-------------: | :--:|
> | Previous SOTA \|  | 47.61|  24.57 | 39.44 | \| |47.52| 18.72|  24.91 |
>
>
>
> >Q3:  The practical implementation of the likelihood score in Eq 4.
>
> A product of many probabilities is indeed a very small number, thus the beam search algorithm  always adopts the following decoding objective:  $P = \frac{1}{n^\beta}\sum_{t=1}^{n} log p(y_t|y_{<t}, x)$  where $n$ is the length and $\beta$ is the length penalty hyperparameter.  Notably,  we practically get the likelihood in Eq 4 by $e^P$ .  The length penalty is set to 1.0 for IWSLT14 De-En, and the likelihood score will be represented as  $\sqrt[n]{\prod_{t=0}^{n} p(y_t |{x}, {{y}}_{<t})}$. As for the example in figure 2, the log likelihood of y' and y'' is respectively -0.7384 ( $e^{-0.7384} \approx 0.48 $) and -0.6369   ($e^{-0.6369} \approx 0.53 $) .  We omitted the details about this for simplicity. We will release our code and add a footnote to explain Eq 4  in our camera-ready version to make it easier to follow.
>
> >Q4:  About sampling methods.
>
> Compared with adopting the sampling method as the reviewer mentioned,  diverse beam search is more similar with the common beam search algorithm that is used in the test stage.
> Based on the reviewer's advice, we have provided the results use top-K/top-P sampling on the summarization benchmark XSum.  Generally, the top-p sampling with temperature $\tau$ set to 0.5 achieves the best result but is still slightly worse than diverse beam search.
>
> | Methods | R-1 | R-2 | R-L |
> | :-----| :----: | :----: | :----: |
> | top-k sampling,  k=50 | 39.29 | 16.74 | 31.67 |
> | top-p sampling,  p=0.6 | 39.17 | 16.56 | 31.44 |
> | top-p sampling p=0.8 | 39.02 | 16.35 | 31.22 |
> | top-p,  p=0.8 ,  $\tau$=0.5 | 39.40 | 16.77 | 31.74 |
> | diverse beam search| **39.66**  | **16.96**  | **31.86** |
>
> >Q6: More notions and explanation for figure 2.
>
> We have polished the caption of figure 2 based on the helpful comments.  We futher explained that z_y’ and z_y’’ are the two feature representations of  hypotheses y' , y'' from beam search,  the double arrow between the feature representations means pair-wise comparison and the lines used to connect z_y and z_x means cosine similarity.

---

### Official Review · Reviewer_gYyK · 2022-07-11

**Rating:** 7
**Confidence:** 4
**Soundness:** 3 good
**Presentation:** 3 good
**Contribution:** 3 good

**Summary:**

This paper proposes to improve text generation with a sequence-level contrastive loss. The loss take advantage of self-generated negatives and consider pairwise ranking relations. The paper also proposes to use the ranking score to rerank the results during inference. Experiments on five generation tasks with ten benchmarks show its effectivenss.

**Questions:**

I am most curious about the computation of the sequence-level representations (i.e., z_x and z_y). This paper spends little efforts in explaining it. Essentially, the contrastive loss is at the sequence-level and the ordinary generative loss is at the token-level. and the key is how to effectively combine them. I am wondering if you test any variants of the computation of sequence-level representations (e.g., different pooling methods or more sophisticated designs).

**Strengths And Weaknesses:**

Strengths:
1. The method is well-motivated. (using self-negatives alleviates the exposure bias problem; pairwise ranking relations address the problem in previous contrastive learning approaches: ignoring the difference between negative samples)
2. The experiments are elaborate. (five generation tasks and detailed ablation and analysis)
3. The results seem good.

Weaknesses:
1. The improvement over Naive CL seems incremental <= as many ideas (self-generated negatives & pairwise loss) are well-known, though in different areas other than text generation.

---

> ### Author Response · Authors · 2022-07-31
> **Response to Reviewer gYyK**
>
> Thanks for your specific feedback.  We are grateful for the acknowledgment that our method is well-motivated. We conducted extensive experiments to show the effectiveness and generalization ability of the method. Moreover, most popular MLE based text generation applications/frameworks can be effortlessly converted to CoNT without modifying model structures.
>
> ----
> > About the computation of sequence-level representations
>
> This is a great question.  For the computation of the sequence-level representations we follow the design of previous work[1] by adopting an average pooling over the encoder/decoder output along the sequence length axis. We also try max-pooling and stochastic pooling and it doesn't affect the performance significantly (Results on IWSLT14 De-En are shown in the following table). We then tend to think computation of sequence representation may not be a main bottleneck of previous contrastive learning frameworks. However,  we admit we only conduct limited experiments with the variants of the computation of sequence-level representations. It is a good idea to further improve the results via exploring more in this direction.
> | Method | BLEU |
> | :-----| ----: |
> |Mean pooling| 35.55 |
> |Max pooling | 35.16 |
> |Stochastic pooling | 35.32 |
>
> [1] Lee et al., Contrastive Learning with Adversarial Perturbations for Conditional Text Generation. ICLR 2021.

---

### Official Review · Reviewer_2ngF · 2022-07-11

**Rating:** 6
**Confidence:** 4
**Soundness:** 3 good
**Presentation:** 3 good
**Contribution:** 2 fair

**Summary:**

The authors propose a new Contrastive Neural Text generation framework CONT, which improves contrastive learning in neural text generation from three aspects: the construction of contrastive examples, the choice of the contrastive loss, and the strategy in decoding. Experimental results on five generation tasks with ten benchmarks show the effectiveness of the proposed framework.

**Questions:**

N/A

**Strengths And Weaknesses:**

Strengths
- The authors propose a simple and effective framework to improve performance of text generation. The framework addresses bottlenecks that prevent contrastive learning from three aspects.
- The authors conduct extensive experiments. Experimental results show that CONT outperforms previous contrastive learning-based methods on ten benchmarks.

Weaknesses
- The use of self-generated texts as negative samples has been used in many works that use reinforcement learning to mitigate exposure bias problem, so the innovation of this strategy is limited. For example,
Self-Critical Sequence Training for Image Captioning,
Discourse-Aware Neural Rewards for Coherent Text Generation.
- The use of additional similarity constraints in decoding is also used by many works.
Considering the above two points, I think the innovation of the whole method is limited.

---

> ### Author Response · Authors · 2022-07-31
> **Response to Reviewer 2ngF**
>
> Thank you for a detailed review.  The reviewer appreciates our simple and effective approach,  and the main concern may lie in the self-generated texts that have also been used in early work (reinforcement learning based generation) and the similarity constraints in decoding as well.
>
> **Answer**:  Firstly, we agree that many training paradigms call inference as a sub-module in the training stage in order to improve consistency.  For example,  reinforcement learning as the reviewer mentioned, aiming to optimize a non-differentiable reward. It's also used in language GAN[1]  to jointly train a discriminator and  glancing training [3] to perform curriculum learning.
>
> To the best of our knowledge,  though incorporating self-generated hypotheses  has been used in text generation. CoNT is the first to leverage the self-generated negatives, which are hard to distinguish from the ground truth, to learn better representations under the contrastive learning framework. Different from reinforcement learning where the self-generated samples are used to calculate a reward,  the self-generated samples with high likelihoods (ensured by the sampling algorithm beam search) but low BLEU score serve as hard negatives in our contrastive method and are pushed away from the ground truth by the pair-wise loss (see our T-SNE experiments, the comparison between figure 4(b, c) ).  In addition, reinforcement learning in text generation usually exhibits the unstable training problem[3] while the contrastive loss of CoNT declines smoothly and stably on all 10 benchmarks and is not susceptible to parameter tuning.
>
> On the other hand, similarity constraint is an intuitive idea. With the contrastive loss, generated samples with higher text similarity (which can be measured by the evaluation metric such as BLEU and ROUGE) with the ground truth has a closer distance to the source sequence which can act as an additional score to alleviate the one of the main disadvantage of beam search -- it can not perform globally scoring.  There are also other work such as post-generation re-ranking systems.  As discussed in our related work section, post-generation re-ranking always model the global score using similarity constraints with another model, which trades simplicity and  efficiency for accuracy. In contrastive neural text generation, we instead use a global decoding objective with the formulation of the contrastive loss and incorporating it into beam search is proven to be a natural and effective idea.
>
>
> [1] Lamprier et al., Generative Cooperative Networks for Natural Language Generation.  ICML 2022.
>
> [2] Qian et al., Glancing Transformer for Non-Autoregressive Neural Machine Translation. ACL 2021.
>
> [3] Choshen et al., On the Weaknesses of Reinforcement Learning for Neural Machine Translation.  ICLR 2020.

---

### Meta-Review · Area_Chair_5pmo · 2022-08-26

**Recommendation:** Accept
**Confidence:** Less certain

**Metareview:**

This paper received mostly positive ratings, but the reviewers also had some overall reservations about the novelty of this work and that makes this paper relatively borderline.

Pros:
- The paper makes three targeted contributions to contrastive learning in text generation to help mitigate exposure bias problem, and the approach of the paper outperforms SoTA contrastive learning methods.
- The work is supported by extensive experimentation, and on a wide variety of tasks (MT, summarization, code comment generation, data-to-text generation and commonsense generation).
- Experimentation appears to be solid for the most part, but reviewers expressed some minor concerns (e.g., about the summarization evaluation)

Cons:
- The reviewers’ main concern is with the relatively limited novelty of the work, as many of its ideas (e.g., self-generated negatives and pairwise loss) are well-known. That said, the application of these methods to text generation appears to be novel.
- The work contains no human evaluation, but automated evaluation is defensible for some of the tasks (MT and summarization at least).

In sum, the work is quite solid and suffers from no major flaws, but the ideas underlying the methods of the paper are not particularly surprising.

**Award:**

No

---

### Decision · Program_Chairs · 2022-09-14

Accept